# Normalization Layer Per-Example Gradients are Sufficient to Predict Gradient Noise Scale in Transformers

**Gavia Gray**
Cerebras Systems
Toronto, Canada
gavia.gray@cerebras.net

**Aman Tiwari**
Subjective
London, UK

**Shane Bergsma**
Cerebras Systems
Toronto, Canada

**Joel Hestness**
Cerebras Systems
Sunnyvale, CA

## Abstract

Per-example gradient norms are a vital ingredient for estimating gradient noise scale (GNS) with minimal variance. Observing the tensor contractions required to compute them, we propose a method with minimal FLOPs in 3D or greater tensor regimes by simultaneously computing the norms while computing the parameter gradients. Using this method we are able to observe the GNS of different layers at higher accuracy than previously possible. We find that the total GNS of contemporary transformer models is predicted well by the GNS of only the normalization layers. As a result, focusing only on the normalization layer, we develop a custom kernel to compute the per-example gradient norms while performing the Layer-Norm backward pass with zero throughput overhead. Tracking GNS on only those layers, we are able to guide a practical batch size schedule that reduces training time by 18% on a Chinchilla-optimal language model.

## 1 Introduction

The gradients gathered during the backward pass while training a neural network are typically inspected via their Frobenius norm, the magnitude of the vector. This gradient vector may be viewed as the sum of gradients computed over each individual example in the minibatch. Each of these has its own norm. In this work, we develop a method to access these norms that works at any scale, for three common layer types in deep learning models: linear, normalization and embedding layers.

One primary application of a per-example gradient norm is in estimating the Gradient Noise Scale (GNS) [39], a metric that has been shown to be useful in training large scale models [9]. The uncertainty of the GNS estimator depends directly on the size of the batch used to compute the small batch gradient norm as shown in Section 2.1. So, the most precise estimate of the GNS is obtained by computing the gradient norms for *each* example in the minibatch: the per-example gradient norm.

To demonstrate GNS measurement in practice we perform experiments on contemporary language model architectures, providing a detailed visualisation of the movement of the GNS components throughout training, presented in Section 4. By inspecting these components it was found that the GNS of the model is highly correlated between layer types, which we give an intuition for in Figure 1.

However, the practical utility of measuring GNS with per-example gradient norms is only present if it can be gathered without affecting training time. Focusing on LayerNorm [4] layers, we note the main speed bottleneck is the memory I/O when not implemented as a fused kernel. To demonstrate this, we develop a custom kernel to compute both the backward pass and the per-example gradient norms at the same time. Using this kernel the throughput overhead of gathering the per-example gradient is zero, even outperforming PyTorch's LayerNorm at larger dimensions. We apply this to a practical batch size schedule case study in Section 5.

38th Conference on Neural Information Processing Systems (NeurIPS 2024).

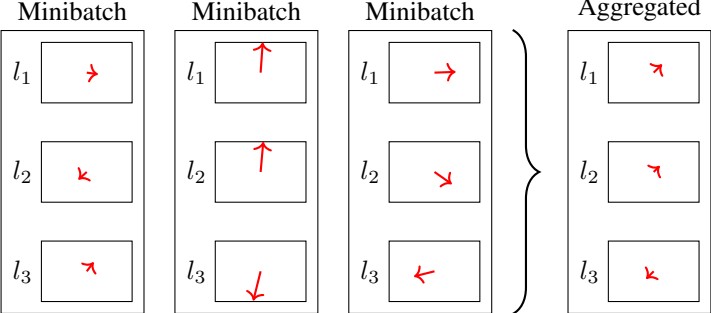

Figure 1: Gradient noise scale (GNS) is typically computed by comparing per-minibatch (aggregated-across-layers) gradients to gradients "Aggregated" across minibatches. We estimate GNS with lower variance by making each minibatch a single example, and maintain per-layer GNS estimates. We find the magnitude of gradients (visualized by the length of red arrows) to be consistent across layers, enabling overall GNS to be computed very cheaply using only gradient stats from LayerNorm layers.

To reiterate, the contributions of this work are:

- A minimal FLOP algorithm and implementation for computing gradients and per-example gradient norms of linear layers simultaneously.[1]
- Observations that the measured GNS for LayerNorm layers is highly correlated with the GNS of the remaining layers.
- Development of an example kernel to implement tracking the GNS of LayerNorm layers that does not affect network throughput (tokens/sec).
- Demonstration of a real application of GNS tracking in a batch size schedule experiment that obtains an 18% wall-time speedup in training a Chinchilla-optimal [29] LLM.

## 2 Background

### 2.1 Gradient Noise Scale

GNS is a metric derived from observing a second order Taylor expansion of the change in a loss function under the following assumption on the noise in the gradient estimate [39],

$$G_{\text{est}(\theta)} \sim \mathcal{N}\left(G(\theta), \frac{1}{B}\Sigma(\theta)\right), \tag{1}$$

where $G_{\text{est}}$ is the observed gradient, $B$ is the batch size, and $\theta$ the parameters of the model. Here, $G$ is the unobserved "true" gradient and $\Sigma$ is the covariance of the gradient estimate. The Taylor expansion mentioned is,

$$\mathbb{E}[L(\theta - \epsilon G_{est})] = L(\theta) - \epsilon|G|^2 + \frac{1}{2}\epsilon^2\left(G^T H G + \frac{tr(H\Sigma)}{B}\right). \tag{2}$$

Where $\epsilon$ is the learning rate and $H$ is the Hessian of the loss. On the right hand side is a factor that depends on $B$. It may be shown [39] that the optimal step size and optimal change in the loss is achieved when $B = \mathcal{B}_{\text{noise}} := tr(H\Sigma)/G^T H G$. Averaging this optimal step over an entire run, and measuring this value by a grid search, yields $\mathcal{B}_{\text{crit}}$ which describes a batch size that meets an optimal tradeoff between cost and training speed. It is shown by analysis and experiment that $\mathcal{B}_{\text{noise}} \approx \mathcal{B}_{\text{crit}}$.

As this depends on the Hessian, which is typically unavailable, McCandlish et al. [39] suggest making the assumption that the Hessian is diagonal, which yields

$$\mathcal{B}_{\text{simple}} = \frac{tr(\Sigma)}{G^T G}. \tag{3}$$

---

[1]Similar algorithms for other layer types described in Appendix B

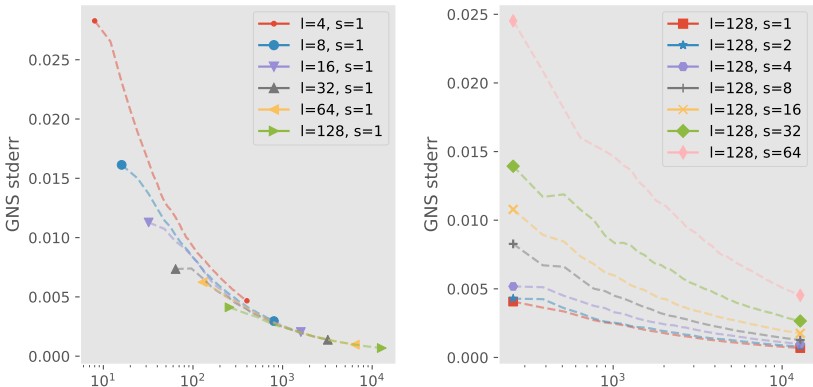

Figure 2: The variance of the GNS estimator for different $B_{\text{big}}$ (left) and $B_{\text{small}}$ (right) sizes. $B_{\text{big}} = l$ and $B_{\text{small}} = s$ in legends. Stderr is estimated using a jackknife resampling method for ratio estimators [12]. For the same number of samples processed, a smaller $B_{\text{small}}$ always has a lower standard error, while the size of the large batch, $B_{\text{big}}$ does not affect the standard error.

To compute $\mathcal{B}_{\text{simple}}$ McCandlish et al. [39] define the unbiased estimators $\mathcal{S}$ and $\|\mathcal{G}\|_2^2$ as:

$$\|\mathcal{G}\|_2^2 := \frac{1}{B_{\text{big}} - B_{\text{small}}} \left( B_{\text{big}} \left\| G_{B_{\text{big}}} \right\|_2^2 - B_{\text{small}} \left\| G_{B_{\text{small}}} \right\|_2^2 \right) \approx G^T G \tag{4}$$

$$\mathcal{S} := \frac{1}{1/B_{\text{small}} - 1/B_{\text{big}}} \left( \left\| G_{B_{\text{small}}} \right\|_2^2 - \left\| G_{B_{\text{big}}} \right\|_2^2 \right) \approx tr(\Sigma), \tag{5}$$

where $B_{\text{big}}$ and $B_{\text{small}}$ are the batch sizes used to compute the gradients $G_{B_{\text{big}}}$ and $G_{B_{\text{small}}}$, respectively (potentially corresponding to *Aggregated* and *Minibatch* gradients as depicted in Figure 1).

$\left\| G_{B_{\text{big}}} \right\|_2$ is trivially computed using the gradients accumulated for the optimizer but $\left\| G_{B_{\text{small}}} \right\|_2$ is not. One option is to use the gradients communicated between Distributed Data Parallel (DDP) nodes, but this has two downsides: (1) the variance of the estimate is tied to the DDP configuration and (2) the estimate is not available in all training configurations. For example, experiments on a single GPU cannot use this method. One can also access the gradients during gradient accumulation, but this similarly depends on the training configuration. A full taxonomy of the options for computing $\left\| G_{B_{\text{small}}} \right\|_2$ is provided in Appendix A.

For each observation of $\left\| G_{B_{\text{big}}} \right\|_2$ we may observe multiple $\left\| G_{B_{\text{small}}} \right\|_2$, typically $B_{\text{big}}/B_{\text{small}}$ of them. On each step the estimate of $\left\| G_{B_{\text{small}}} \right\|_2^2$ is therefore a mean over $B_{\text{big}}/B_{\text{small}}$ samples, whose variance is reduced according to the law of large numbers. However, the GNS is a ratio of the unbiased estimators in Equations 4 and 5, so it may not be clear how this affects uncertainty in the GNS estimate. Figure 2 explores this relationship by simulation of a setting where the GNS is set to 1 while varying $B_{\text{big}}$ and $B_{\text{small}}$. We find it is always better (less uncertainty) to use the smallest possible $B_{\text{small}}$ to estimate the GNS, while the choice of $B_{\text{big}}$ is irrelevant.

## 2.2 Efficient Per-example Gradient Norms

Goodfellow [26] proposes a trick to compute gradient norms for individual examples in a minibatch, which would provide the minimum variance estimate of the GNS as described in Section 2.1. Neglecting the original derivation, by writing the desired squared norm as a tensor contraction the trick may be reproduced automatically via `einsum` path optimization [49, 15]. The tensor contraction for per-example gradient norms, $n_b^2$, of a linear layer in the 2D setting is,

$$n_b^2 = \sum_{i,k} (w')_{bik}^2 = \sum_{i,k} x_{bi} x_{bi} y'_{bk} y'_{bk},$$

where $x$ are the activations prior to a linear layer, $y'$ are the gradients of the loss with respect to the outputs of the linear layer and $w'$ are the gradients of the loss with respect to the weights of the linear layer.

Li et al. [36] extend this trick to the three dimensional case. For inputs $\mathbf{X} \in \mathbb{R}^{B \times T \times I}$ and outputs $\mathbf{Y} \in \mathbb{R}^{B \times T \times K}$, the per-example gradient norm $n_b$ is,

$$n_b^2 = (w')_{bik}^2 = (\sum_t x_{bti} y'_{btk})^2 = x_{bti} y'_{btk} x_{bui} y'_{buk} = \langle \mathbf{X} \mathbf{X}^T, \mathbf{Y}' \mathbf{Y}'^T \rangle_F^2,$$

which has $O(T^2)$ memory complexity in the sequence length $T$.[2] Index sets are $b \in [1, B]$, $i \in [1, I]$, $k \in [1, K]$, $t, u \in [1, T]$. At some point, the I/O cost of computing the per-example gradient norms by computing the full $w'_b$ explicitly will be cheaper. Noting this fact motivated the work in Section 3 and the practical relationship between these resource costs is explored in Section 3.1.

## 2.3 Related Work

**Gradient norms**    One common motivation for computing per-example gradient norms is for differential privacy. By bounding the gradient for any single example, we can ensure each example has a limited impact on the final parameters [45, 36]. Per-example gradient clipping has been performed with convolutional networks [45] and sequential models, e.g., LLMs [36]. These methods allow control over per-example gradient norms even when training with large batch sizes. Approaches like these are implemented in the differential-privacy library Opacus [56], and have support natively in PyTorch, but are less efficient than the methods proposed in this paper. An alternative mechanism to manifest per-example gradient norms is to simply use a batch size of one. While not efficient enough for training large-scale networks, such sequential training may arise in situations such as reinforcement learning, where per-example gradient clipping has also been performed (to improve stability [52]).

**Gradient noise scale**    The Gradient Noise Scale [39] has been widely used for training large-scale neural networks. For example, Brown et al. [9] note the GNS was measured during training and used to guide batch sizing when training GPT-3. Dey et al. [19] mention that operating near the critical batch size, as dictated by the GNS, is important for hyperparameter transfer under the maximal update parameterization [54]. Even when not explicitly mentioned in publications, open source code often implements the GNS (e.g., see codebases [21, 13] for GPT-NeoX [7] and Hourglass Diffusion Transformer [14]).

Measurements similar to the GNS have also been used in a range of prior work to guide batch sizing for minibatch SGD [10, 17, 5, 55]. Chen et al. [11] show experimentally that wider networks can be trained using larger batches; they also establish a theoretical connection between wider networks and gradient variance, albeit for simple two-layer networks. In contrast, Shallue et al. [47] found empirically that *narrower* Transformers scale better to larger batch sizes. Smith and Le [50] propose a noise scale based not on gradient variance, but on the learning rate, dataset size, and batch size (similar to the notion of temperature in Section 4.1). Zhang et al. [60] find the critical batch size depends on the choice of optimizer. Faghri et al. [22] introduce a gradient clustering and stratified sampling approach to minimize minibatch gradient variance, and use this approach as a tool to help understand optimization.

**Gradient variance**    Beyond computing the GNS, our method can support other applications where measuring the distribution of per-example gradients is useful or informative. Gradient variance has been used to classify the *difficulty* of examples [1], which can be used, for example, to surface problematic examples for human auditing. The question of whether gradient distributions tend toward Gaussian in the (central) limit is of theoretical significance [50], with implications toward the ability of SGD to escape sharp minima and land in wide basins [63, 41, 48]. Bounded gradient variance is also assumed in some convergence analysis [8, 62], as noted in [22].

Perhaps the most familiar use of gradient variance is of course in adaptive optimizers like Adagrad, Adam, and others that reduce step sizes in high-gradient-noise directions [20, 57, 46, 33, 44]. Hilton et al. [28, App. C] directly relate Adam second moment statistics to a *component-wise* version of the GNS. Optimizers typically estimate gradients jointly across training steps and minibatches, however vSGD [46] leverages separate components for gradient momentum and for gradient variation across samples. Zhang et al. [61] find the variance of gradient norms across examples predictive of whether

---

[2]This specific Einstein contraction is not used by Li et al. [36] but appears in the Backpack library [15] We provide the vector algebra contraction path chosen by Li et al. [36] on the right.

vanilla SGD outperforms adaptive optimizers, however recent work has shown Adam to outperform SGD even in the (noise-free) full gradient descent setting [34, 35].

# 3 Simultaneous Per-example Gradient Norms

As described in Section 2, computing GNS requires small batch gradient norms. Typically, these may be gathered during gradient accumulation or DDP communication.[3] However, these methods are not universally applicable and may not be available in all training configurations. In this section we describe a method for baking the computation of the per-example gradient norms into the computation graph, making it universally applicable. The typical tensor contraction used to compute the backward gradient in a linear layer using the input activations, $\mathbf{x}$, and gradients, $\mathbf{g}$, is,

$$w'_{k,l} = \sum x_{...k} g_{...l},$$

in other words, a sum over vector outer products for every vector in the trailing dimension. In principle, it is possible to access the intermediate tensor containing the batch dimension $w'_{bkl} = \sum x_{b...k} g_{b...l}$. This allows us to compute the per-example gradient norms with FLOPs scaling at the same rate as the normal, non-per-example backward pass (Figure 3), albeit at increased I/O cost due to having to materialize the intermediate tensor.

A generic algorithm to compute the per-example gradient norms simultaneously with the weight gradient in a standard linear layer is provided in Algorithm 1 using `einsum` for readability and portability.[4] The reason for the correction in step 4 can be seen by considering the gradient of loss function $L$ with respect to the weights on a single example $b$, $w_b$,

$$\nabla_{w_b} \frac{1}{B} \sum_b L(x_b) = \frac{1}{B} \nabla_{w_b} L(x_b),$$

computing the squared norm of this will therefore contain a factor of $1/B^2$, which must be corrected for.

---

**Algorithm 1** Linear Layer Simultaneous Per-Example Gradient Norm Computation

---

**Require:** gradient tensor $\mathbf{g}$ of shape $(B, ..., L)$, input activation tensor $\mathbf{x}$ of shape $(B, ..., K)$
**Ensure:** weight gradient tensor $\mathbf{w}'$ of $(K, L)$, mean of per-example squared norms $\|\mathbf{w}'_b\|_2^2$
  1: $\mathbf{w}'_b \leftarrow \text{einsum}('b...k, b...l \rightarrow bkl', \mathbf{x}, \mathbf{g})$
  2: $\mathbf{s}_w \leftarrow \text{einsum}('bkl \rightarrow b', \mathbf{w}'^2_b)$
  3: $\mathbf{w}' \leftarrow \text{einsum}('bkl \rightarrow kl', \mathbf{w}'_b)$
  4: $\|\mathbf{w}'_b\|_2^2 \leftarrow 1/B \times \text{einsum}(\mathbf{s}_w, 'b \rightarrow ') \times B^2$ # reduce by mean then apply correction
  5: **return** $\mathbf{w}', \|\mathbf{w}'_b\|_2^2$

---

## 3.1 FLOPs and I/O Costs

The computational cost of computing per-example gradient norms can be broken down into FLOPs, in Figure 3, and I/O, in Figure 4, with matrix multiplication on current devices being potentially bottlenecked by both. We estimate ideal FLOP and DRAM I/O costs, assuming optimal reuse of data loaded from DRAM into SRAM with no recomputation. In practice, duplicate computation may be used to improve wall-clock time and to fit within hardware limitations of the amount of shared memory available. We compare here against the efficient per-example gradient norm method described by Li et al. [36], which the authors note is only efficient (in terms of I/O cost) when $2T^2 < PD$, where $T$ is the sequence length, $P$ is input and $D$ is output dimension of the linear layer. This bound is discussed further in Appendix E.

In terms of FLOPS, Figure 3 shows the simultaneous per-example gradient norms are almost always preferable, only being more expensive for very short sequence lengths in small models. The reason for this is shown on the right hand side; the number of FLOPs required to compute the simultaneous per-example gradient norms is independent of the sequence length.

---

[3]A complete taxonomy for small batch gradient computation is given in Appendix A.
[4]Additional algorithms for Embedding and LayerNorm layers are described in Appendix B.

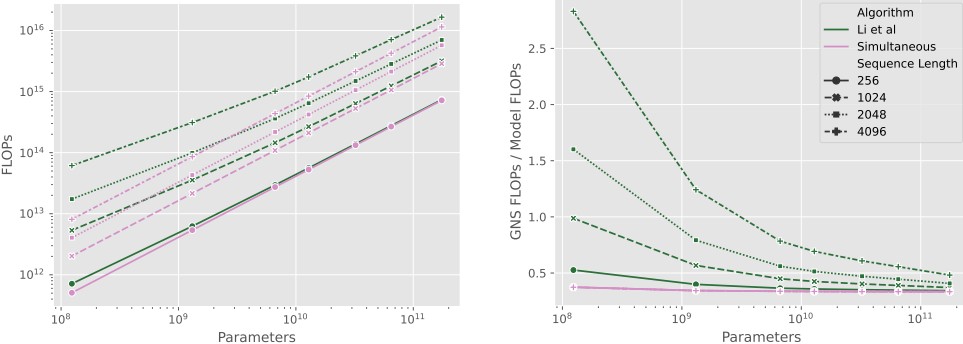

Figure 3: FLOP cost of computing per-example gradient norms. (Left) Total FLOP cost. (Right) Proportional cost versus one model forward and backward pass. The FLOP cost of Simultaneous per-example gradient norms is strictly dominant to alternative methods (left) and the ratio of this additional cost to the FLOP cost of processing the entire model does not depend on context length (right).

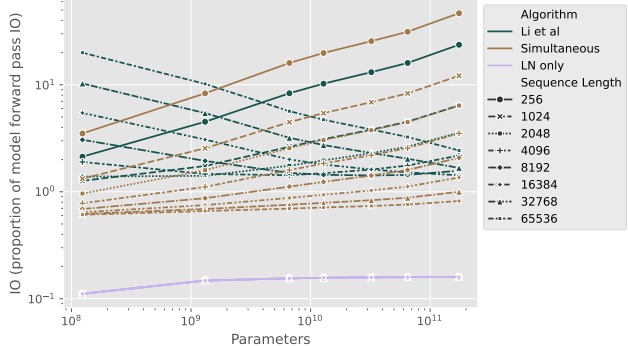

Figure 4: Total I/O cost of computing per-example gradient norms, assuming gradients and parameters are stored with 4 bytes of precision. The relative IO cost of Simultaneous per-example gradient norms is less than Li et al. [36] for very long contexts for all model scales, approximately equivalent for models of 10B parameters and 4096 context length, and higher for shorter contexts with larger models. The IO cost of LN (LayerNorm) per-example gradient norms alone is much lower than either method.

The I/O cost shown in 4 illustrates a tradeoff in computing the per-example gradient norm. The simultaneous method is more expensive at large model sizes with short sequence length because it must act on a large intermediate tensor.

To estimate model flops, we use PyTorch's FLOPCounterMode, which only measures the FLOPs in matrix multiplications and attention computation, however these make up the vast majority of the FLOPs in a Transformer model.

## 4    Gradient Noise Scale in Transformer Language Models

Using the methods described in previous sections to measure per-example gradient norms and estimate the GNS, we perform experiments on a 111M parameter Chinchilla-optimal language model [19, 29] using the OpenWebText dataset [24].[5] As the prior work was performed on Pile [23], Appendix C.1 describes an experiment to check the optimality of the Chinchilla model on this dataset. We also found Flash attention led to numerical instability, which we were able to mitigate with an architectural modification described in Appendix C.2.

[5]The code to replicate these experiments may be found at https://github.com/CerebrasResearch/nanoGNS/tree/main/exact.

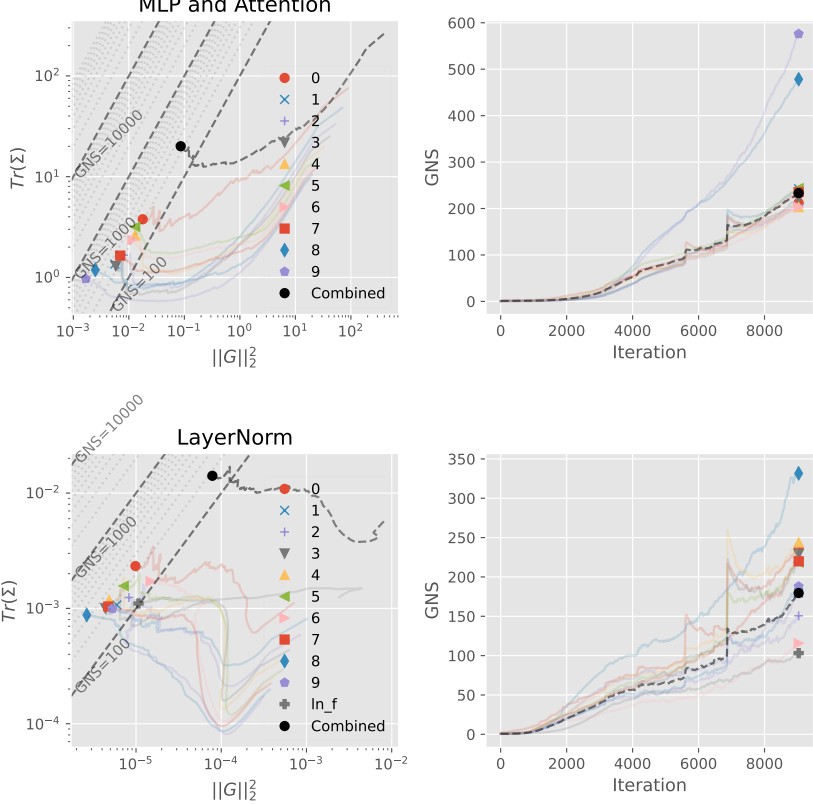

Figure 5: GNS phase plot: Linear/Embedding layers are separated from LayerNorm layers by row. Component estimators of Equations 4 and 5 are shown (left) with the GNS over the course of training on the (right).

All experiments computed per-example gradient norms for all layers in the model with the exception of the performance results of Sections 5.1 and 5.2, which only computed per-example gradient norms for the normalization layers. Each experiment was run on Nvidia A10 GPUs, in either 12 or 24 hours depending on the precision used, Bfloat16 or Float32 respectively. We used the nanoGPT[6] codebase with the layers described in Section 3 added.

Having an accurate estimate of the GNS statistics $\|\mathcal{G}\|_2^2$ and $\mathcal{S}$ allows us to visualize the movement of both in a phase space during training as shown in Figure 5. LayerNorm layers are separate from the rest of the network because their statistics are much smaller and to illustrate how the resulting GNS estimates on the right track each other. To observe these trends in another training regime, see Figure 14 in Appendix D.1.

## 4.1 The Temperature of Training

McCandlish et al. [39, App. C] observed that the GNS measurement depends on the batch size and learning rate used in training. In fact, from the derivation outlined in Section 2.1, the gradient noise scale is only well-defined at the optimal learning rate. Using a toy model of a quadratic loss function, they observed that the GNS should be inversely proportional to the temperature, $T$, a ratio of batch size $B$ to learning rate $\epsilon$:

$$\mathcal{B}_{\text{noise}} \propto \mathcal{B}_{\text{simple}} \propto \frac{1}{T} = \frac{B}{\epsilon}.$$

This enables a testable prediction that the GNS will increase with increasing batch size or with descending learning rate. This prediction was found to accurately describe experiments on a small

---

[6]https://github.com/karpathy/nanoGPT

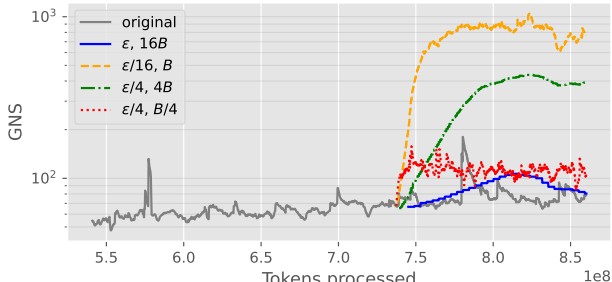

Figure 6: During the middle of training a 111M parameter language model on OpenWebText, the learning rate, $\epsilon$ or batch size, $B$ were varied, restarting the run from the same point. This Figure replicates an experiment from McCandlish et al. [39] showing how varying the ratio causes changes in the measured GNS, but here only due to changes in the learning rate. Changes in the batch size do not have the predicted effect.

convolutional model on the SVHN dataset. We repeat it here in the setting described above in Figure 6. To match the results of McCandlish et al. [39], all interventions tested should yield the same result. We find the GNS does indeed react predictably to changes in the learning rate, but the reactions to changes in the batch size are not predicted by the theory.

### 4.2 GNS Correlates Between Layer Types

Inspection of Figure 5 suggests the LayerNorm layers produce a similar GNS, when combined, as the total GNS of the model. Before describing how to quantify this relationship we must first note that the unbiased estimators $\|\mathcal{G}\|_2^2$ and $\mathcal{S}$ are noisy. All GNS figures presented in this paper and other work smooth both of these estimators, typically with an Exponential Moving Average (EMA) filter, before computing the GNS ratio.[7]

So, when quantifying the relationship between the GNS of different layers, it must be compared for different smoothing factors. Here, we show the regression coefficients with respect to the alpha of the EMA filter in Figure 7. The results show that the GNS of the LayerNorm and Attention layers are highly predictive of the total GNS of the model. In both cases, the slope is approximately 1.4, meaning the total GNS is approximately 1.4 times the GNS of the LayerNorm or Attention layers.

Comparing the quality of this fit versus the quality of prior work's overall fit of the GNS to the critical batch size (measured empirically) [39], the quality seems acceptable and we do not need to apply this 1.4x correction factor, rather we just note that the true $\mathcal{B}_{\text{crit}}$ may be greater than the measured $\mathcal{B}_{\text{simple}}$.

## 5 Batch Size Scheduling

We focus on two concerns that affect the practicality of batch size scheduling. First, measuring the appropriate batch size without incurring any additional training time. We find this is possible with the method described in Section 5.1. Second, whether batch size scheduling is effective in practice. We find it can offer significant savings in the required number of tokens processed in Section 5.2.

### 5.1 Universal GNS with Zero Overhead

Capturing a GNS estimate for a linear layer is powerful, but efficiently doing so presents a challenge. Such an estimate requires accumulating per-example gradients of hidden_size$^2$ across the sequence dimension, compared to just hidden_size with LayerNorm. This increased size requires using more complex reductions in the kernel, rather than a simple warp reduction followed by shared-memory atomic reduction with a final atomic global reduction (as we can implement for LayerNorm per-example gradients within shared memory). In addition, linear layer kernels are already highly optimized and require using advanced techniques to keep GPU tensor cores fed with data, so

---

[7]The results described in Figure 5 are explored at a 1.3B parameter scale in Appendix D.3.

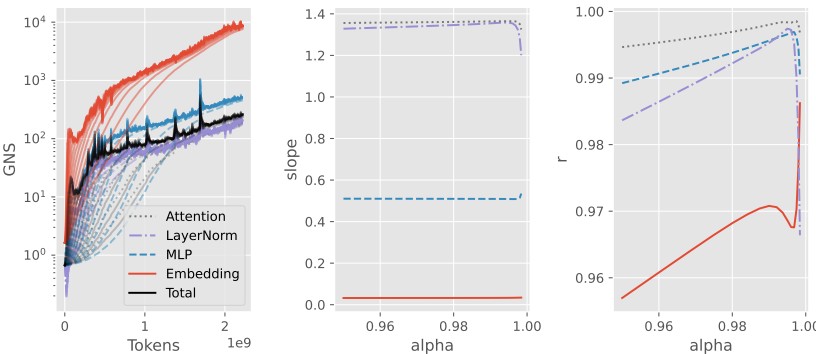

Figure 7: Regression of total GNS using the GNS of each layer type. (Left) GNS of each layer type and the total GNS are plotted against the number of tokens processed for varying EMA alpha settings. (Center & Right) The slope and Pearson's correlation coefficient of the regression of the total GNS against the GNS of each layer type, respectively, as a function of the same EMA alpha values. The total GNS (black) on the left is predicted well by individual layer types as indicated by the correlation coefficients (right), however the type with slope closest to 1 is LayerNorm (center), only overestimating the GNS by less than 40% across EMA alpha values.

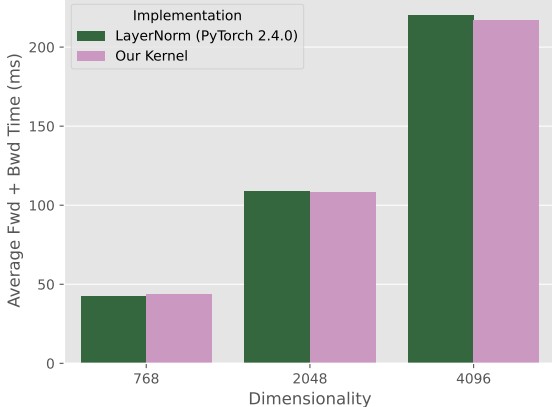

Figure 8: Comparison of average time taken for a LayerNorm forward and backward pass with gradient accumulation when using PyTorch's native implementation versus our custom kernel computing per-example gradient norms in tandem. Measured on an Nvidia H100 GPU.

combining such a kernel with per-example gradient computation - with its own memory overheads and corresponding available bandwidth reduction - would be a difficult undertaking.

We thus implemented a LayerNorm-specific CUDA kernel that also captures GNS. In experiments with language models at different scales, illustrated in Figure 8, we find this kernel has practically zero overhead compared to PyTorch's LayerNorm implementation. The complete source code for this kernel is provided with the accompanying code for this paper[8].

## 5.2 Case Study: Batch Size Schedule

As a case study we continue with the 111M parameter language model on OpenWebText described above. Over three seeds, we run both a fixed batch size and a batch size schedule that increases linearly with the number of tokens processed to the original batch size. We vary the batch size during training by varying the number of gradient accumulation steps.

---

[8]https://github.com/CerebrasResearch/nanoGNS/tree/main/exact/normgnorm

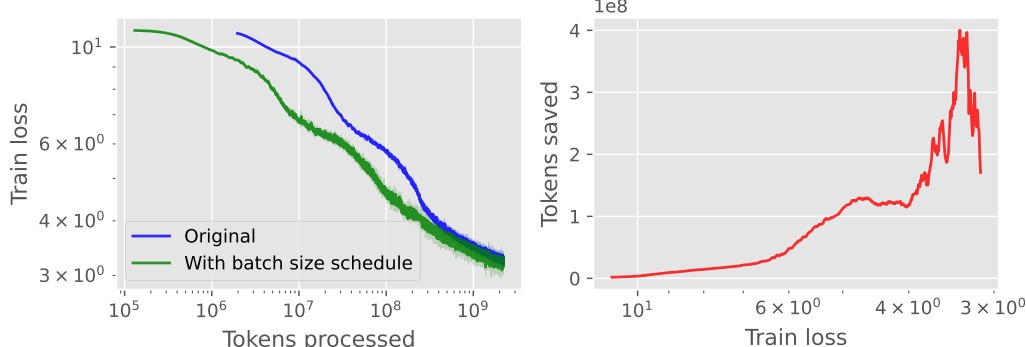

Figure 9: (Left) Linear batch size schedule tracking the GNS over 2.2 billion tokens processed. Loss is plotted over a smoothed range from 3 runs using different Seeds. (Right) The number of tokens saved over the fixed batch size run to achieve the same loss.

The results of this experiment are shown in Figure 9. The left plot shows the progression of the loss for both models, with the range of values captured over different seeds. The mean loss for the linear batch size schedule leads the fixed batch size throughout training. On the right, this lead is quantified by interpolating the number of tokens saved to achieve the same loss. The precise schedule used is shown in Figure 15 in Appendix D.2.

# 6 Limitations

In this paper, we only studied Transformers, which include Normalization sub-layers natively. While Transformers are ubiquitous in machine learning, there are many models, including variations of RNNs, CNNs, and state-space models, that do not use such layers conventionally. However, we note LayerNorm could be added to these networks with very little overhead (in fact, the desire to normalize activations in RNNs was one of the original motivations for developing LayerNorm; application of batch normalization [30] to RNNs was "not obvious" [4]). Nevertheless, investigating LayerNorm-based GNS in these other models requires further work.

Our work is also part of efforts to improve efficiency and address the increasing costs of training and tuning large neural networks [6]. We provide both a more-efficient technique for computing the GNS, and also, by enabling use of GNS statistics, we support compute-efficient training recipes, such as use of dynamic batch sizes. While some have argued that hyperscalers may re-invest any efficiency savings into ever-larger models [42], for academic researchers, such savings could allow pushing the state-of-the-art, while still getting results in a reasonable timeframe. Recent efforts to enable frontier-model-performance within academic budgets are encouraging, both to reduce memory [38, 18] and save compute [37, 2]. Of course, even for such economical approaches, "extensive hyperparameter search" may still be required [31]. There is a growing awareness that hyperparameter tuning has a negative impact on equity in AI research, as tuning success depends directly on researcher finances [51]. A correlated trend is to use better training measurements (such as gradient noise in batch and step size optimizers (Section 2.3)) to reduce dependence on hyperparameters, and in this way we hope our work can also ultimately improve research equity.

# 7 Conclusion

This work set out to provide a practical method for computing the per-example gradient norms necessary to compute the GNS independent of the training configuration. In the process we discovered that not all the layers are necessary for a practical estimate of the GNS and that the per-example gradient norms can be computed for the normalization layers with zero overhead. This enabled practical experiments, such as a batch size schedule and replicating prior GNS observations. We are hopeful that democratising access to GNS statistics, on any device, will enable subsequent discoveries.

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

# A  Taxonomy

Gray et al. [27] included an prior version of this taxonomy in their work.

The following taxonomy describes the different methods available to compute the $\|G_{B_{\text{small}}}\|_2^2$ necessary to compute the GNS as described in Section 2.1. "Gradient norm cost" below refers to the cost of computing the norm of the gradient for all parameters in the model, which is typically orders of magnitude smaller than the cost of forward or backward passes.

- Microbatch: multiple $\|G_{B_{\text{small}}}\|_2^2$ are computed over a set of microbatches
    - DDP: Each $\|G_{B_{\text{small}}}\|_2^2$ is computed before gradients are communicated between DDP nodes [39].
      Pros: Only gradient norm cost.
      Cons: Variance tied to number of DDP nodes (see Figure 2), can't be used on one node.
    - Sequential: Each $\|G_{B_{\text{small}}}\|_2^2$ are computed sequentially during gradient accumulation.
      Pros: Only gradient norm cost.
      Cons: Variance tied to the number of gradient accumulation steps.
- Subbatch: During gradient accumulation, select $\|G_{B_{\text{small}}}\|_2^2$ partway through.
  Pros: Only gradient norm cost, easy to implement.
  Cons: Higher variance than Microbatch as $\|G_{B_{\text{small}}}\|_2^2$ is not averaged.
- Per-example:
  Pros: Independent of gradient accumulation or DDP configuration, minimal variance.
    - Exact:
        * $\|G_{B_{\text{small}}}\|_2^2$ is computed directly by the per-example gradient trick [26, 36].
          Pros: Minimal cost in 2D regime.
          Cons: Redundant computation required in 3D regime.
        * $\|G_{B_{\text{small}}}\|_2^2$ is computed in tandem with the parameter gradients using the method described in Section 3.
          Pros: No redundant computation.
          Cons: Expansion in memory causes slowdowns as described in Section 3.1.
    - Approximation: $\|G_{B_{\text{small}}}\|_2^2$ is approximated by assuming input activations are normally distributed with mean zero [27].
      Pros: Fewer FLOPs than Exact methods.
      Cons: Not exact.

All of the methods described above can be measured either online or offline. The description above focuses on the online case; i.e. measuring the gradient norms during training. To use these methods offline: run the models without performing weight updates and measure gradient norms the same way. The estimators of Equation 4 and 5 can then be aggregated using a mean rather than an EMA or by using a method to estimate measurement uncertainty such as the jackknife mentioned in Figure 2 (described in the context of GNS by Gray et al. [27, App.B]). This can be useful to estimate how long to run the offline estimate.

# B  Additional Simultaneous Per-Example Gradient Norm Computations

Algorithms 3 and 2 describe the process for computing the per-example gradient norms for the embedding and LayerNorm layers, which are typically the remaining layers in Transformer models. RMSNorm [59] is practically identical to LayerNorm in this case because the parameters the gradient is computed wrt are in the affine transform, which is the same in both layer types.

# C  Language Model Experiment Details

As mentioned in the text, the code to run the experiments described in this paper can be found at https://github.com/CerebrasResearch/nanoGNS/tree/main/exact.

---
**Algorithm 2** Layernorm Simultaneous Per-Example Gradient Norm Computation
---
**Require:** gradient tensor $\mathbf{g}$ of shape $(B, ..., K)$, input activation tensor $\mathbf{x}$ of shape $(B, ..., K)$
**Ensure:** gamma gradient tensor $\boldsymbol{\gamma}'$ of shape $(K,)$, mean of per-example squared norms $\|\boldsymbol{\gamma}'_b\|_2^2$,
    gradient tensor $\boldsymbol{\beta}'$ of shape $(K,)$, mean of per-example squared norms $\|\boldsymbol{\beta}'_b\|_2^2$
  1: $\boldsymbol{\gamma}'_b \leftarrow$ einsum('$b...k, b...k \rightarrow bk$', $\mathbf{x}, \mathbf{g}$)
  2: $\mathbf{s}_\gamma \leftarrow$ einsum('$bk \rightarrow b$', $\boldsymbol{\gamma}'^2_b$)
  3: $\boldsymbol{\gamma}' \leftarrow$ einsum('$bk \rightarrow k$', $\boldsymbol{\gamma}'_b$)
  4: $\|\boldsymbol{\gamma}'_b\|_2^2 \leftarrow 1/B \times$ einsum($\mathbf{s}_\gamma$, '$b \rightarrow$ ') $\times B^2$ # reduce by mean then apply correction
  5: $\boldsymbol{\beta}'_b \leftarrow$ einsum('$b...k \rightarrow bk$', $\mathbf{g}$)
  6: $\mathbf{s}_\beta \leftarrow$ einsum('$bk \rightarrow b$', $\boldsymbol{\beta}'^2_b$)
  7: $\boldsymbol{\beta}' \leftarrow$ einsum('$bk \rightarrow k$', $\boldsymbol{\beta}'_b$)
  8: $\|\boldsymbol{\beta}'_b\|_2^2 \leftarrow 1/B \times$ einsum($\mathbf{s}_\beta$, '$b \rightarrow$ ') $\times B^2$ # reduce by mean then apply correction
  9: **return** $\boldsymbol{\gamma}', \|\boldsymbol{\gamma}'_b\|_2^2, \boldsymbol{\beta}', \|\boldsymbol{\beta}'_b\|_2^2$
---

---
**Algorithm 3** Embedding Layer Simultaneous Per-Example Gradient Norm Computation
---
**Require:** gradient tensor $\mathbf{g}$ of shape $(B, T, D)$, input id tensor $\mathbf{x}$ of shape $(B, T)$, vocabulary size $V$
**Ensure:** weight gradient tensor $\mathbf{w}'$ of shape $(V, D)$, mean of per-example squared norms $\|\mathbf{w}'_b\|_2^2$
  1: $\mathbf{o} \leftarrow$ onehot($\mathbf{x}, V$)
  2: $\mathbf{w}'_b \leftarrow$ einsum('$btv, btd \rightarrow bvd$', $\mathbf{o}, \mathbf{g}$)
  3: $\mathbf{s}_w \leftarrow$ einsum('$bvd \rightarrow b$', $\mathbf{w}'^2_b$)
  4: $\mathbf{w}' \leftarrow$ einsum('$bvd \rightarrow vd$', $\mathbf{w}'_b$)
  5: $\|\mathbf{w}'_b\|_2^2 \leftarrow 1/B \times$ einsum($\mathbf{s}_w$, '$b \rightarrow$ ') $\times B^2$ # reduce by mean then apply correction
  6: **return** $\mathbf{w}', \|\mathbf{w}'_b\|_2^2$
---

## C.1 Optimality on OpenWebText

We chose to use the Cerebras-GPT [19] recipes for experiments as they are designed to be Chinchilla optimal. This means that each model size should achieve the lowest possible loss for a given FLOP budget [29]. However, these recipes were tuned on the Pile dataset [23] and we used the OpenWebText dataset [24] so that results could be replicated (Pile is no longer publicly available).

To verify that the training protocol is optimal on OpenWebText, we performed a small study to illustrate how the performance would vary as we vary the size and total tokens trained on. Model size was varied by changing the hidden size: the 70M model has a hidden size of 576, the 111M model has a hidden size of 768 and the 161M model has a hidden size of 960. The token budget for each model size was chosen to keep the total FLOPs constant.

The learning rate was varied to observe a minima in the loss at each model scale. The results are shown in Figure 10. While we found that the learning rate may be increased overall, the 111M model was found to have the lowest loss of the three models. From these results we conclude that the training protocol is optimal within this range of model sizes and we assume 111M is good enough. In other words, a better model might exist between 70M and 161M parameters for this FLOP budget but it isn't outside of this range.

## C.2 Flash Attention Numerical Instability

The experiments described in Sections 4 and 5.2 involve Chinchilla optimal language models at a 111M scale [19]. These experiments were replicated according to the published information. We encountered diverging runs when executing in bfloat16 Automatic Mixed Precision (AMP) consistent with the default settings in nanoGPT. These experiments were executed on NVIDIA A10 GPUs for accessible replication at small scale. By ablation it was found that these runs would diverge:

    • Regardless of batch size schedule

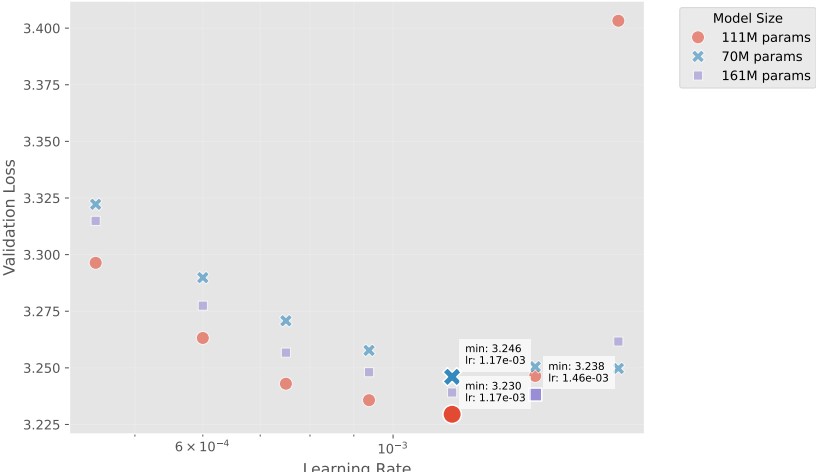

Figure 10: The loss of models trained on OpenWebText with 70M, 111M and 161M parameters. The learning rate was varied to find the minima in the loss at each model scale. The optimal learning rate for each model size is annotated.

- Regardless of hyperparameters: learning rate, weight decay, LayerNorm epsilon or Adam epsilon [53]
- When using PyTorch's AMP in bfloat16 precision
- When using Flash attention [16, 25]

This was surprising because prior work had trained these models successfully [19]. In that work the model was also trained using bfloat16 AMP precision, but it was trained on a Cerebras CS-2 system. Due to this difference, we suspected the issue was due to a difference between the efficient attention kernel and the Flash attention kernel in PyTorch.

By inspecting the histograms of weights and biases in the Query, Key, Value (QKV) projection during training, we found that range grew the fastest in block 1 (the *second* block in the model). In addition, we observed that the histogram of the query and key projection weights became *bimodal* as the gradient norm diverged. This is illustrated in Figure 11. Further analysis of a checkpoint taken at this point in training focused on the difference between gradients computed using the flash attention kernel and the nanoGPT pure PyTorch attention implementation using float32 precision. At initialization the gradients were not significantly different but at the point of divergence there was a significant difference coinciding with increased parameter norms in that layer.

To replicate the issue from scratch, we came up with a simulation from a generic initialization. Inspired by the teacher-student experiment protocol proposed by Ba et al. [3] (although otherwise unrelated) we set up a learning task with a "teacher" and "student" model with the same architecture. Both networks begin with the same weights but we add a small amount of noise to the teacher's weights. The student is trained to match the teacher's output. After experimenting with hyperparameters we were able to replicated the divergence seen during training[9], as illustrated in Figure 12.

Using this isolated simulation we were able to test different methods to mitigate the divergence. Karras et al. [32] suggested that cosine attention could address similar divergences attributed to self-attention. In Figure 13 we replicated the experiment described in Figure 12 using cosine attention and found that the divergence no longer occurred.

Separately, experimenting with precision ablation found that if float32 precision was used only in block 1 (2nd) then the divergence would also not occur. Based on this and the above, we found the following two architectural mitigations for the divergence, in *only block 1 (2nd)*:

- Use cosine attention, i.e. normalize the query and key head vectors before self-attention. *OR*

---

[9]The code for this experiment is available at `https://gist.github.com/gaviag-cerebras/b77aef9de29e859a5e999a582d57f6a2`

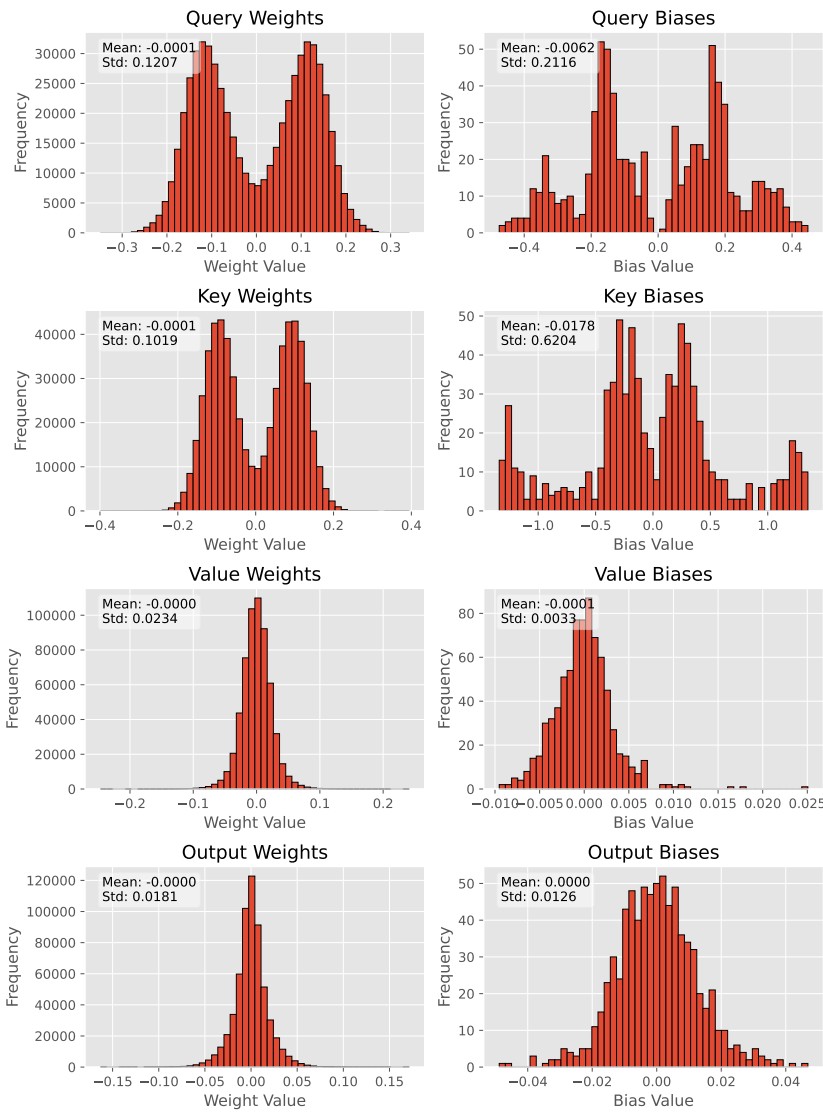

Figure 11: Histograms of weights and biases for the 111M experiment described in Sections 4 and 5.2 from the attention block containing the QKV projection, self-attention and output projection layers. The histograms for the query and key projection weights and biases are bimodal while the value projection weights and biases are not.

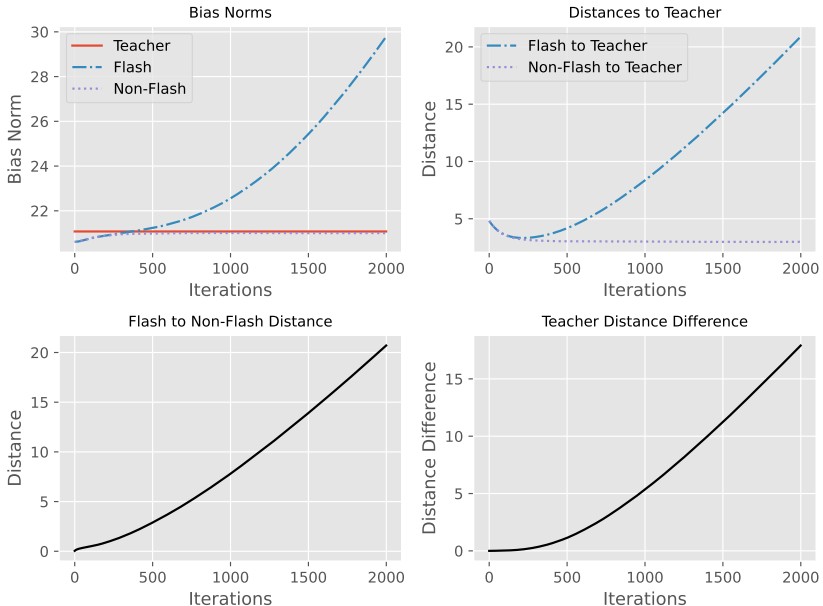

Figure 12: Two "student" networks, identical to the "teacher" network except for the addition of a small amount of noise to the teacher's QKV projection bias. As training progresses, the student using Flash attention diverges for the same inputs. Plots are, clockwise from top left: "Bias Norms" shows the norms of the bias layer in each of the networks, "Distances to Teacher" shows the L2 distance from each student to the teacher. "Flash to Non-Flash Distance" shows the L2 distance between the student using Flash attention and not, "Teacher Distance Difference" is the difference between the distances to the teacher for both cases.

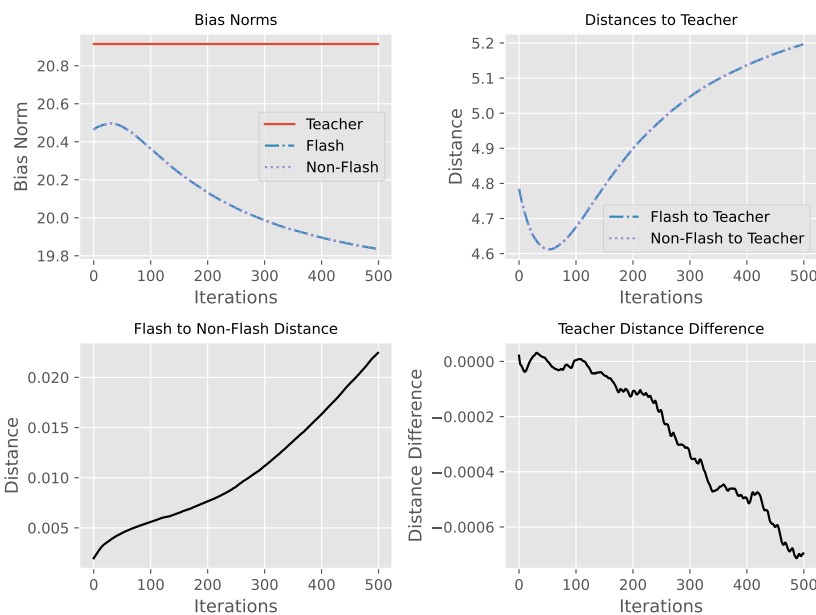

Figure 13: Replication of the experiment described in Figure 12 using cosine attention instead of Flash attention. The divergence observed no longer occurs.

- Use spectral normalization [40] on the QKV projection.

Critically, only modifying a single layer does not affect the throughput of the model, the observed Model FLOPs Utilization (MFU) did not decrease by more than 1% in either case. Both of these bound the norm of the query and key head vectors prior to attention. Spectral normalization achieves this because the QKV projection is preceded by a LayerNorm layer. Using this mitigation on the 111M model allowed the experiment to be replicated on an NVIDIA A10 GPU and we observed the same behaviour as running the model more slowly in float32 precision.

Similar divergences are discussed in prior literature (and in nanoGPT's issue tracker) but we are unable to verify that it is the same problem. Wortsman et al. [53] discuss how to build similar experiments to those described above but do not investigate flash attention specifically. Golden et al. [25] investigate the numerical stability of Flash attention but neglect to demonstrate a failure mode that affects real training runs. Zhai et al. [58] focus on the numerical stability of attention in general and propose a similar mitigation (their method, $\sigma$Reparam, is a scaled version of spectral normalization) but do not investigate flash attention specifically.

It is likely that the mitigation proposed will not work in all cases, such as for larger models. However, we only needed to replicate at the scale we were working at. The experiments in Figure 12 and Figure 13 are included to illustrate how bounding the norm of the query and key head vectors seems to be important for numerical stability. However, this may change in future versions of the flash attention kernel, these results were obtained with PyTorch 2.4.0.

# D  Additional GNS Results

## D.1  Additional GNS Phase Plot

Figure 14 shows the GNS phase plot for the same model as described in Section 4 but with the linear batch size schedule described in Section 5.2.

## D.2  Batch Size Schedule

The batch size schedule used in the experiment described in Section 5.2 is shown in Figure 15.

## D.3  Larger Scale Training

To demonstrate that the method scales to larger models, we trained a 1.3B parameter GPT model[10] on OpenWebText using 8 H100 GPUs. The results of this experiment are shown in Figure 16. The left plot shows the per-example gradient norms for all layers, while the right plot shows the per-example gradient norms for only the LayerNorm layers. The GNS computed using the traditional DDP method is also shown for comparison. In Figure 16a we observe that the LayerNorm remains predictive of the total GNS, as in the 111M model results of Figure 7. When all non-fused simultaneous per-example gradient norms were collected we observed an MFU of 40% and when only the fused LayerNorm layers were collected we observed an MFU of 57%.

After completing this experiment a bug was discovered in the code that decreased per-example gradient norms by a constant factor. This caused an underestimation of the GNS. In Figure 16b this can be seen when we compare the GNS estimated via DDP method. Initially, we assumed that this constant factor was due a failure of the LayerNorm GNS approximation to larger models. Unfortunately, we did not have the budget in time or resources to rerun the experiment so we corrected the results by multiplying by the constant factor observed in the comparison to the DDP method.

This may be representative of real world scenarios where a large model is pretrained over many DDP nodes. As the user has access to two methods to estimate the GNS, they may account for any bias or slope between the estimates. Then, if it is necessary to continue training on a single node, they can use the per-example gradient norms to estimate the GNS. Similar techniques can involve enabling per-example GNS estimation for all layers for a short time, or estimating the GNS offline as described in Appendix A.

---

[10]Again following the GPT2-like[43] prescription from Dey et al. [19].

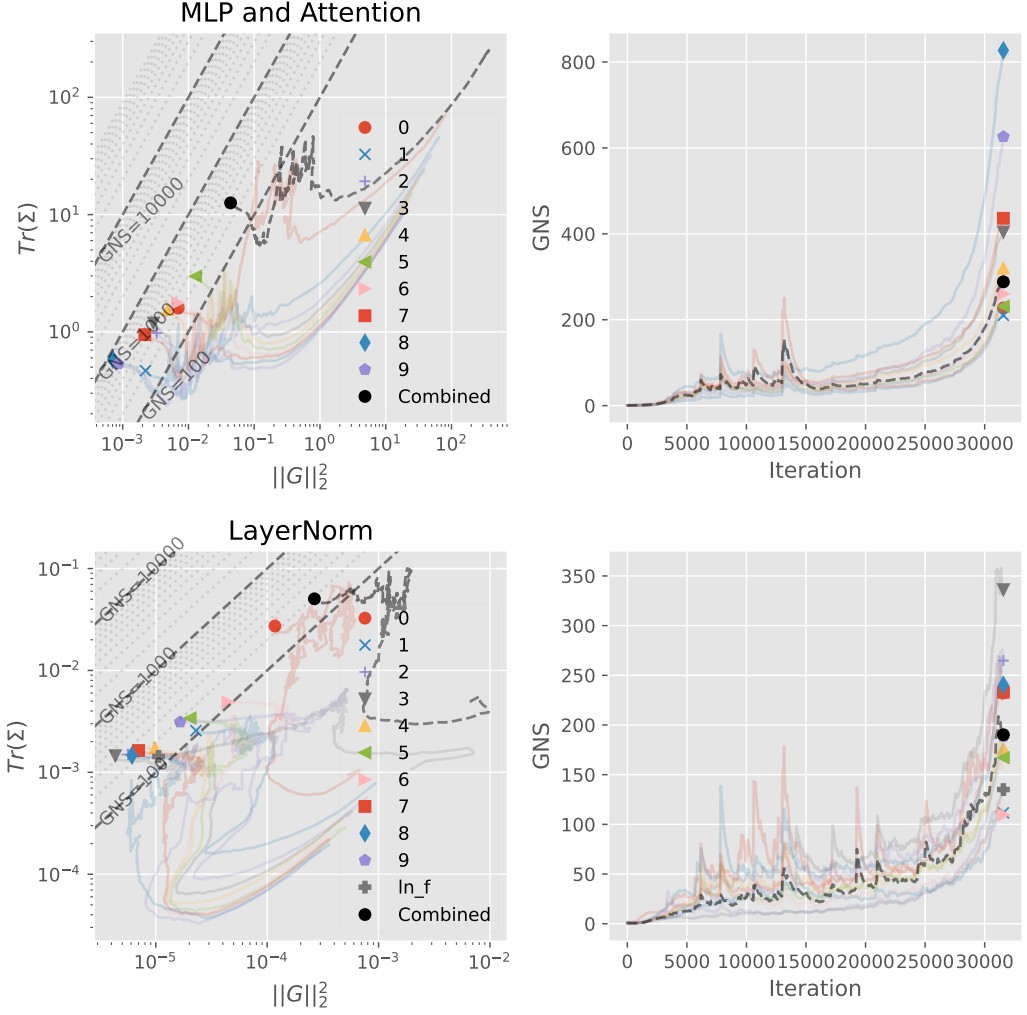

Figure 14: GNS phase plot as in Figure 5 but focusing on the batch size schedule described in Section 5.2. Linear/Embedding layers are separated from LayerNorm layers by row and the component estimators of Equations 4 and 5 are plotted (left), with the GNS over the course of training (right).

# E   FLOP & I/O Formulae

We use the following formulae in our FLOP and I/O cost estimations, where $B =$ Batch Size, $T =$ Sequence Length, $K =$ Input Dimension, $L =$ Output Dimension:

Table 1: FLOPs

| Algorithm | Weight Gradient | Gradient Norms |
|---|---|---|
| Simultaneous | $BKL\,(2T-1) + KL\,(B-1)$ | $BKL + B\,(KL-1)$ |
| [36] | $KL\,(2BT-1)$ | $BT^2 \cdot (2K + 2L - 2) + BT^2$ |

Table 2: I/O

| Algorithm | Weight Gradient | Gradient Norms |
|---|---|---|
| Simultaneous | $BKL + BKT + BLT$ | $BKL + B$ |
| [36] | $BKT + BLT + KL$ | $2BT^2 + B$ |

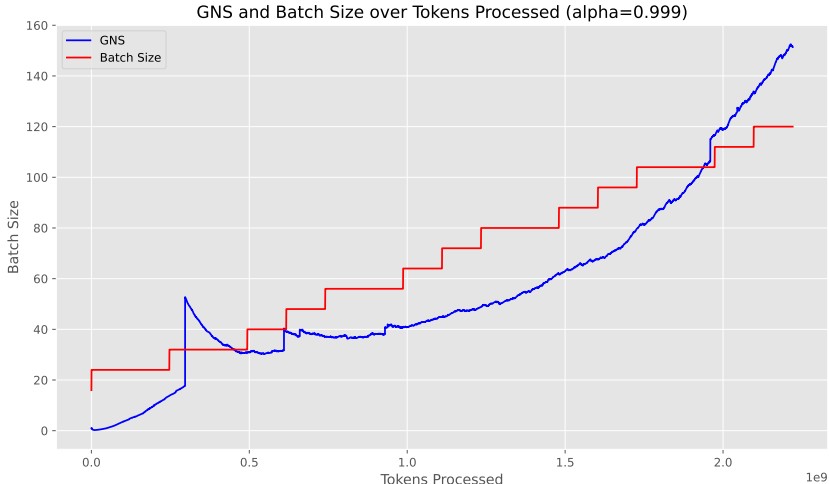

Figure 15: The batch size schedule used and GNS observed in the 111M batch size schedule experiment illustrated in Figure 15. An aliasing issue is noticeable in the interpolated linear batch size schedule that was used. This has since been fixed in the published code.

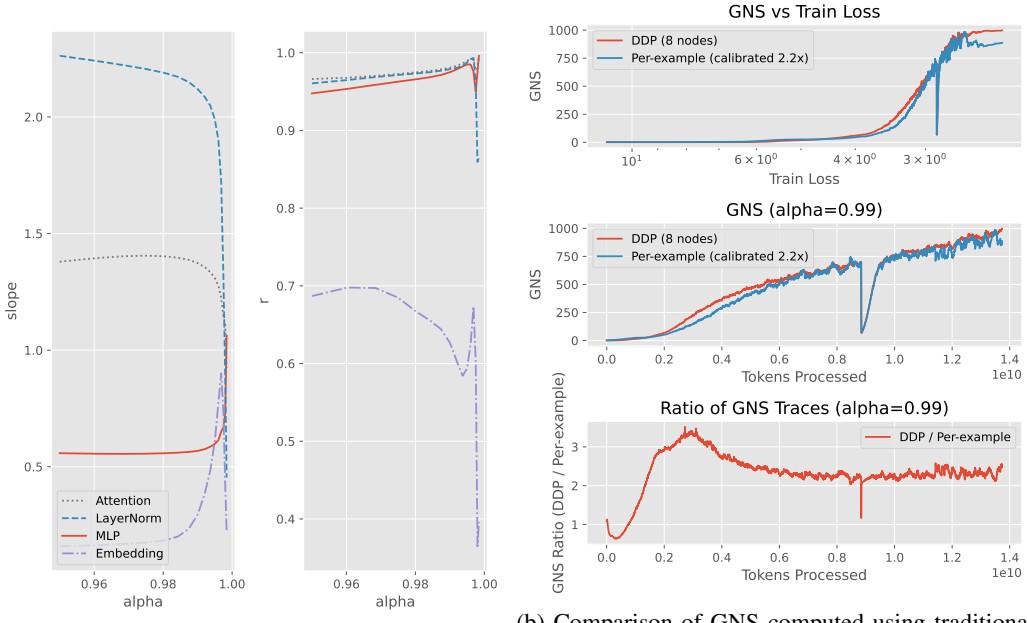

(a) Regression analysis repeating Figure 6.

(b) Comparison of GNS computed using traditional DDP methods and per-example gradient norms.

Figure 16: 1.3B GPT model train on OpenWebText using 8 H100s, trained twice. (Left) Per-example gradient norms for all layers were gathered to replicate the analysis in Figure 7. (Right) Per-example gradient norms were gathered for only LayerNorm layers, then compared to the GNS computed using traditional DDP methods.

Solving the I/O equations above reproduces [36]'s analysis with $T = \frac{\sqrt{2}\sqrt{KL}}{2}$ at the cross-over point above which simultaneous calculation is more I/O efficient. Solving for FLOPs gives:

$$T = \sqrt{\frac{2KL - 1}{2K + 2L - 1}}.$$

