# OpenReview forum: "Normalization Layer Per-Example Gradients are Sufficient to Predict Gradient Noise Scale in Transformers"
_NeurIPS.cc/2024/Conference — NeurIPS 2024 poster_

### Official Review · Reviewer_32m2 · 2024-07-04

**Soundness:** 2
**Presentation:** 1
**Contribution:** 3
**Rating:** 4
**Confidence:** 4

**Summary:**

This paper explores efficient ways of computing the gradient noise scale (GNS) when training transformers. The main practical relevance of the GNS is that it can be used to estimate the critical batch size, where the larger batch sizes become computationally inefficient. The authors discuss different ways of doing this at a fine granularity both time and layerwise. They show that the backwards pass through a layernorm can be modified to estimate their GNS for essentially free and that this predicts the GNS of other layers. The authors then vary the batch size throughout training based on this, showing this can potentially save compute for a given target model performance.

**Strengths:**

- The core idea of estimating the GNS in a cheap way based on normalization layers is interesting and relevant to the community.
- The proposed layernorm based estimation can be performed very cheaply.

**Weaknesses:**

- Some more work is required to make the proposed method practical due to numerical issues. Although the idea is interesting it would be much more impactful if the kernel worked as a drop-in replacement.
- The 18% speed improvement claimed in the abstract may be an overclaim due to the numerical issues and presumably only being applicable to certain types of training setups (maybe one GPU doing gradient accumulation rather than distributed setups).
- The paper is quite hard to follow, despite the relatively simple ideas presented.
- The use of Einstein notation contributes to this, it is fine for brevity but not clarity.
- The paper relies too heavily on McCandlish et al 2018. Despite spending significant effort on trying to summarize the relevant portions of this paper, many things are still unclear in the later sections.
- The figures are hard to interpret, both due to very short captions that do not summarize the high level idea and takeaway, and insufficient labels on the figures themselves.

**Questions:**

Overall I think the core ideas are interesting but this paper would strongly benefit from being resubmitted with improved writing and overall presentation.

Suggestions for improvement:
- Drop the Einstein notation, make things explicit instead of implicit.
- Consider rewriting the introduction to gently introduce the GNS, maybe with a diagram and summarize the high level importance of this before diving in. What are the high level ideas needed to understand your contribution? Do you really need to go into all these details upfront?
- Improve the description of your figures. Take Figure 1 for example, it is very hard to parse. The caption does not summarize the message, it should also emphasize how the dashed and plotted lines are related.
- For Figures 2 and 3, maybe emphasize more what Li et al is, what the differences are and what you expect to see? Why not normalize the IO costs as well in the comparison (maybe you need some assumptions on the other kernels. Without some sort of reference number it is hard to see how much these IO overheads matter in practice.
- Figure 5: Again, maybe tell us what is expected? What is the takeaway?
- Figure 6: It is very unclear what is going on in the two plots on the right. Almost no description is given in the text or the caption of the “EMA alpha values”.
- Batch Size Schedule: It is not clear how “a batch size schedule that increases linearly with the number of tokens processed to the original batch size” relates to your GNS estimations. Do you set the length of the schedule based on the GNS? Is this done based on prior estimates of the GNS or in an online fashion?
- Figure 8: This should probably be done using learning rate schedules of different lengths for statements like 2x to be meaningful.

**Limitations:**

Some limitations discussed, negative societal impact not applicable

---

> ### Author Rebuttal · Authors · 2024-08-07
>
> Thank you so much for your detailed and constructive review, and your kind words regarding the appeal and relevance of the paper’s core idea.
>
> ### On the clarity of the paper
>
> > The paper relies too heavily on McCandlish et al 2018. Despite spending significant effort on trying to summarize the relevant portions of this paper, many things are still unclear in the later sections.
>
> Unfortunately, despite the known use of GNS as detailed in the Related Work
> section, there is limited research published on the topic since McCandlish et al.
> (2018). However, GNS estimation is of practical value in our work and we believe
> it will be for the community.
>
> ### On Einstein notation
>
> > Drop the Einstein notation, make things explicit instead of implicit.
>
> We agree that Einstein notation may be unclear. The most straightforward way to
> address this problem would be to write out the sums explicitly. However, we find
> the utility of the Einstein notation to be the ability to express that the order
> of the summation is flexible. Another better solution would be to write example
> matrix operations. For the equation on line 75, we could have written
> $$
> W'_b = X_b {Y'}_b^T, \quad n_b = ||W'_b||_F^2
> $$
> as one possible contraction. We will add these examples to the paper to make the
> notation more clear.
>
> > Consider rewriting the introduction to gently introduce the GNS, maybe with a diagram and summarize the high level importance of this before diving in. What are the high level ideas needed to understand your contribution? Do you really need to go into all these details upfront?
>
> Thanks for this suggestion. Based on this we have created a diagram to shown in
> our attached pdf in Figure 1. We think this may help prime the reader for the
> concepts in the paper.
>
> ### On Figure messaging
>
> > The figures are hard to interpret, both due to very short captions that do not summarize the high level idea and takeaway, and insufficient labels on the figures themselves.
>
> > Improve the description of your figures. Take Figure 1 for example, it is very hard to parse. The caption does not summarize the message, it should also emphasize how the dashed and plotted lines are related.
>
> Thank you for this feedback. We agree that the figure captions should state the
> message of the figures. For example, in Figure 1, we will say, "For the same
> number of samples processed, a smaller $B_{small}$ always has a lower standard
> error (dashed), while the size of the large batch $B_{large}$ does not affect
> the standard error." The solid lines in this figure are the estimate of the GNS
> at each number of samples processed, supposed to illustrate the uncertainty
> caused by the finite number of samples. To simplify the figure it will be
> clearer to remove the solid lines.
>
> > For Figures 2 and 3, maybe emphasize more what Li et al is, what the differences are and what you expect to see? Why not normalize the IO costs as well in the comparison?
>
> We agree that it would be beneficial to provide more context for Figures 2 and 3.
> We will add the following messages to the captions:
>
> - Figure 2: "...The FLOP cost of Simultaneous per-example gradient norms is strictly dominant to alternative methods (left) and the ratio of this additional cost to the FLOP cost of processing the entire model does not depend on context length (right).
> - Figure 3: "...The I/O cost of Simultaneous per-example norms is greater for models of 10B parameters or more but approximately equivalent for models of 1B parameters or less, depending on context length."
>
> In addition we have tested normalized I/O costs in the Figure and found that it
> makes that Figure more clear. We will use this Figure in the updated paper.
>
> > Figure 5: Again, maybe tell us what is expected? What is the takeaway?
>
> To Figure 5 we will add the message, "This Figure replicates an experiment from
> McCandlish et al. (2018) showing how the ratio of $\epsilon / B$ causes changes
> in the measured GNS but only due to changes in the learning rate. The batch size
> does not have the predicted effect."
>
> > Figure 6: It is very unclear what is going on in the two plots on the right. Almost no description is given in the text or the caption of the “EMA alpha values”.
>
> We thank the reviewer for noticing that the description of the EMA on lines
> 177-178 does not mention how the alpha value controls smoothing. We will add an
> Appendix making the role of the alpha value and add EMA pseudo code, because
> there are many ways to implement it.
>
> For Figure 6 we will add the message, "The total GNS (black) on the left is
> predicted well by individual layer types as indicated by the correlation
> coefficients (right), however the type with slope closest to 1 is LayerNorm
> (center),
> only overestimating the GNS by less than 40% across EMA alpha values."
>
> ### On batch size scheduling
>
> > Batch Size Schedule: ... Do you set the length of the schedule based on the GNS? Is this done based on prior estimates of the GNS or in an online fashion?
>
> This is a great question! The batch size schedule we used linearly increased
> during training because we had observed this trend in the GNS and this is
> illustrated in Figure 3 of our attached pdf. We agree that an automatic online
> batch size schedule using GNS would be ideal but we didn't want to
> overcomplicate the paper. The schedule is supposed to mimic what a practitioner
> watching the GNS might do; in practice an operator adjusts the batch size
> manually for expensive runs.
>
> > Figure 8: This should probably be done using learning rate schedules of different lengths for statements like 2x to be meaningful.
>
> We agree that varying hyperparameters would make this result more robust. At
> this scale, we observed that the batch size scheduled run was outperforming
> a tuned baseline, so we did not pursue this further

---

> > ### Comment · Reviewer_32m2 · 2024-08-11
> >
> > I thank the authors for their response and clarifications. It is very hard to evaluate how the proposed changes would affect the clarity of the paper, which is its largest weakness. I think the authors may be on the right track, but a thorough evaluation of the changes requires reviewing the paper again in its new form which can only be done at a different venue / cycle. However, as the authors address the numerical and speed issues to some extent, I will raise my score to 4.
> >
> > A few followups from the rebuttal:
> > * Regarding the speedup, I still believe this will depend on your setup, including whether you do local accumulation or DDP. The total batch size amortizes other costs such as the communication and the local batch size is important for per-device utilization. With gradient accumulation on a single GPU you don't really have communication costs and you can keep the microbatch the same during your batch size scheduling. With DDP you either have to decrease the microbatch size or reduce the number of workers which has additional overheads.
> > * I still really encourage you to reconsider the Einstein notation. I asked a couple of my colleagues about this and they agree that using it will significantly limit the readability and accessibility of a paper. This will of course differ between sub-communities, but I believe many people in the field are still not comfortable with it.
> > * For your rebuttal Figure 1: "We find the magnitude of gradients (visualized by the length of red arrows) to be consistent across
> > layers, enabling overall GNS to be computed very cheaply using only gradient stats from LayerNorm layers." It seems you are assuming the gradients have a very low mean compared to the variance here, otherwise I believe you would have to account for the mean component too, not just the magnitude. Maybe make try to make this explicit somehow.

---

> > > ### Author Response · Authors · 2024-08-12
> > >
> > > Thanks for taking the time to review our comments and raise your score.
> > >
> > > >  I think the authors may be on the right track, but a thorough evaluation of the changes requires reviewing the paper again in its new form which can only be done at a different venue / cycle. However, as the authors address the numerical and speed issues to some extent, I will raise my score to 4.
> > >
> > > While we acknowledge the need for presentational improvements, we want to make it clear that no changes are proposed to the method or key contributions of the paper (i.e., changes that would necessitate a new review cycle).  The attached pdf was intended mainly to clarify a numerical issue tangential to our method.  The proposed changes to the paper itself are quite surgical and totally feasible by the camera-ready deadline: revising figure captions, adding 1.3B model results to appendices, revising the point in related work on adaptive optimizers, editing the introduction to preview contributions, adding [[3][]] to related work, adding pros and cons of per-example gradient norm methods to appendices, explaining $B, I, K$ variables, using `\citet` only in the first instance of citations, explaining big/small batch size definitions, adding vector algebra to the equation on line 75, adding the attached diagram to the introduction, removing solid lines from Figure 1, and adding an illustration of the batch size schedule to Figure 8.  Thanks again to you and the other reviewers for your great suggestions for improving the paper's clarity!
> > >
> > > > Regarding the speedup, I still believe this will depend on your setup, including whether you do local accumulation or DDP. The total batch size amortizes other costs such as the communication and the local batch size is important for per-device utilization. With gradient accumulation on a single GPU you don't really have communication costs and you can keep the microbatch the same during your batch size scheduling...
> > >
> > > To be clear, our method does not depend on gradient accumulation (we can vary the number of gradient accumulation steps and the results are exactly the same for the same global batch size).  The small batch size is *not* the microbatch size used during gradient accumulation; the small batch is never materialised, rather, we use a per-example gradient norm trick that allows us to get the gradient norms for each example as if we ran the experiment with a microbatch size of 1, when in reality we did not.
> > >
> > > > I still really encourage you to reconsider the Einstein notation. I asked a couple of my colleagues about this and they agree that using it will significantly limit the readability and accessibility of a paper. This will of course differ between sub-communities, but I believe many people in the field are still not comfortable with it.
> > >
> > > Thanks again for taking the time to think more on this. We understand that some researchers do not prefer Einstein notation. However, the idea for the method we present was directly inspired by observing the form of Equation on line 75 and this is a common representation. For example, the Backpack library uses this exact same contraction for computing per-example gradient norms (they call them batch l2 norms). For example, see the implementation of convolution [[1][]] and linear layers [[2][]]. These are an equivalent, less efficient, version of what we present.
> > >
> > > We believe that because this representation of the contraction is invariant to the order of summation it is useful as a representation of the numerical problem. The solution (ie reduction path) presented in the proposed algorithms is a solution to that problem.
> > >
> > > Also, we have suggested a vector algebra example of a possible reduction path for this contraction in our original response. Did you have any comment on this? Would you have preferred us to rewrite the einsum with the explicit sums included?
> > >
> > > > For your rebuttal Figure 1: "We find the magnitude of gradients (visualized by the length of red arrows) to be consistent across layers, enabling overall GNS to be computed very cheaply using only gradient stats from LayerNorm layers." It seems you are assuming the gradients have a very low mean compared to the variance here, otherwise I believe you would have to account for the mean component too, not just the magnitude. Maybe make try to make this explicit somehow.
> > >
> > > Thanks for your input on this diagram. The relationship between the mean and variance is not really captured in this simplification, it is intended to prime the reader to think about the norms across layers and across examples in minibatches. The caption will be simplified to express this.
> > >
> > > [1]: https://github.com/f-dangel/backpack/blob/1ebfb4055be72ed9e0f9d101d78806bd4119645e/backpack/extensions/firstorder/batch_l2_grad/convnd.py#L30
> > > [2]: https://github.com/f-dangel/backpack/blob/1ebfb4055be72ed9e0f9d101d78806bd4119645e/backpack/extensions/firstorder/batch_l2_grad/linear.py#L50-L52
> > > [3]: https://arxiv.org/abs/2204.02311

---

### Official Review · Reviewer_fwbc · 2024-07-12

**Soundness:** 3
**Presentation:** 2
**Contribution:** 1
**Rating:** 6
**Confidence:** 4

**Summary:**

The paper proposes a method for efficient computation of per-example gradient norm for the broader usecase of computing gradient noise scale. Further the authors showcase the usecase of gradient noise scale (GNS) in Transformers and showcases that GNS of only normalization layers in transformer models suffices, rather than the total GNS. Using this proposal, the paper proposes a batch size scheduling resulting in training time reduction.

**Strengths:**

The paper for the most part is well written and has shown useful applications of gradient noise scale in Transformer architectures which can be further extrapolated to other architectures. e.g. state space models.

- Related work section discussion is well motivated to discuss the utility of gradient noise scaling, perhaps a more better way to represent this section would have been to segregate the related work with smaller sections with headers - one discussing practical utility and other definitions used in literature for gradient noise scaling and any recent efforts towards efficient computation for per-gradient norms.

**Weaknesses:**

Overall, I feel the main contributions of the paper is not well presented in this draft and in many portions the writing is hard to follow and seems disconnected from other sections in the paper.

On one aspect the authors propose existing methods of efficient per-example gradient norms in Section 2.2 but only a few papers like Li et al. [29] and Goodfellow [22] are mostly discussed. It would be nice to have a much more better discussion of more related work to have a thorough pros and cons discussion.

Notation writing could be improved to a better extent.
For instance in Sec 2.2, line 74 which space does B, I, K indicate or is just a placeholder for 3D tensor dimensions.
Similarily line 75 $x_{bti}y^{'}_{btk}x_{bui}y^{'}_{buk}$ , how does the index indicate u ?

Related work discussion on other methods on per-gradient norm efficient computation is missing. For instance https://openreview.net/forum?id=xINTMAvPQA is cited but this one or similar other approaches how they have approached this problem and how the authors proposal differs in that respect , that thorough discussion would be appreciated.

**Minor fixes in Writing**

- Section 2.1: it may be shown [32] -> It may be shown McCandlish et al. [32].  And you may remove McClandish elsewhere.
- Section 2.1 : I do not see any specific discussion on $B_{big}$ and $B_{small}$. How do they differ? Any references to later section in the paper where they define the size differences.

**Questions:**

Step 4. of Algorithm 1. (what is the exact correction) and is very limited in scope in terms of contribution.

**Limitations:**

Yes, I appreciate that the authors have adequately addressed the limitations of this work in terms of their empiricial evaluations only on Transformer architectures and not on other similar architectures like RNNs or state-space models, which is possibly left as a future work. Apart from this there are no limitations or any potential negative social impact of their work.

---

> ### Author Rebuttal · Authors · 2024-08-07
>
> Thank you for your helpful review.  Your suggestions will definitely help improve the paper.  It is also gratifying to know that the work is well motivated and the application of per-example gradient norms to Transformers training is useful and clear.
>
> ### Regarding revising the related work
>
> >  perhaps a better way to represent this section would have been to segregate the related work with smaller sections with headers
>
> We thank the reviewer for this suggestion. We will add `\paragraph` headers to the related work section to make it easier to navigate.
>
> ### Regarding clarifying the contributions
>
> > Overall, I feel the main contributions of the paper is not well presented in this draft
>
> We plan to address this suggestion through a new figure in Section 1 that
> highlights the unique contributions of our method.  For example, using something
> like Rebuttal PDF Figure 1 and the figure caption, we will contrast our
> (layerwise) per-example norms with other methods that must aggregate gradients
> at a coarser level based on their specific data-parallel configuration.  We will
> also do a better job of highlighting the specific contibutions in the
> introduction, previewing the experimental investiation that unfolds over the
> rest of the paper.
>
> We will also do a better job of motivating the use of per-example gradient norm
> applications beyond GNS estimation (such as for differential privacy).  We
> should have also made it more clear that GNS itself is rarely used because it
> traditionally requires large DDP setups to be useful. With our method, it can be
> applied both on the smallest MNIST experiment or extremely large scale language
> model training.
>
> ### Regarding per-example norms in related work
>
> > only a few papers like Li et al. [29] and Goodfellow [22] are discussed … it would be nice to have … more related work to have a thorough pros and cons discussion.
>
> This is a good suggestion, thanks.  While related work in this area is limited,
> following your suggestion, we did discover ["Efficient Per-Example Gradient
> Computations in Convolutional Neural
> Networks"][pe_conv](). This paper should have been
> mentioned in the related work section and we will add it in the final version.
> As [backpack][] makes clear, this also reduces to a 3D tensor regime, so will be
> equivalent to our method. Li et al. [29] is the only method we know of that
> focuses directly on per-example norms in 3D tensor regimes, which motivated our
> comparison in Section 3 (lines 111-148). We agree that a discussion on pros and
> cons would be valuable, for example Goodfellow [22] is optimal in 2D tensor
> regimes, and we will add this in the final work.  Thanks again!
>
> [backpack]: https://github.com/f-dangel/backpack
>
> > discussion on other methods on per-gradient norm efficient computation is missing. For instance https://openreview.net/forum?id=xINTMAvPQA is cited but this one or similar other approaches how they have approached this problem and how the authors proposal differs in that respect
>
> We agree that a thorough discussion of this comparison would be valuable. At
> present the paper only provides a short comparison in Appendix A (line 389).
> Appendix A currently only compares the methods in the context of GNS estimation.
> To improve it, we will list pros and cons of all known per-example gradient norm
> estimation methods [22, 29, [rochette2019efficient][pe_conv],
> [gray2023efficient][]] directly in the final version and discuss their usage in
> GNS estimation.
>
> [pe_conv]: https://arxiv.org/abs/2204.02311
> [gray2023efficient]: https://openreview.net/forum?id=xINTMAvPQA
>
> ### Regarding notation and citations
>
> > Notation writing could be improved … for instance in Sec 2.2, line 74 which space does B, I, K indicate or is just a placeholder for 3D tensor dimensions. Similarily line 75 $x_{bti}{y^{'}}_{btk}x_{bui}{y^{'}}_{buk}$ , how does the index indicate u ?
>
> We agree that the interpretation of $B$, $I$, and $K$ should be included. We
> will add notes that they correspond to the batch size, input dimension, and
> output dimension, respectively. Additionally, the $u$ index shares the same
> index space $u \in (1, ..., T)$ as $t$, which may cause confusion. We will
> clarify this in the final version.  Good catch!
>
> > Section 2.1: it may be shown [32] -> It may be shown McCandlish et al. [32]. And you may remove McClandish elsewhere.
>
> We thanks the reviewer for pointing this out, we mechanically applied `\citet`
> without considering how many times "McCandlish et al." would get printed. We
> will address this in the final version.
>
> > Section 2.1 : I do not see any specific discussion on $B_{big}$ and $B_{small}$. How do they differ? Any references to later section in the paper where they define the size differences.
>
> We thank the reviewer for pointing out this gap. On line 50 $B_{big}$ and
> $B_{small}$ have an incomplete definition. In the final version we will explain
> that $B_{big}$ is typically the full batch size and $B_{small}$ is a fraction of
> that, typically the batch size executed on DDP nodes or during gradient
> accumulation.
>
> ### Regarding Algorithm 1
>
> > Step 4. of Algorithm 1. (what is the exact correction) and is very limited in scope in terms of contribution.
>
> We will expand the discussion in the paper to make the contribution of line
> 4 clear. We agree that the correction on line 4 of Algorithm 1 is only
> a consequence of backpropagation of losses that have been reduced by a mean; it
> is not strictly part of the per-example gradient norm estimation. Lines 123-128
> describe why this correction exists and we found that it was often a source of
> error in new implementations.
>
> The form of the correction in Algorithm 1 is because it is safer to apply the
> correction to the norm after taking the mean of the squared norm $s_w$ (it would
> be equivalent to sum the loss and then scale the update gradients, but this is
> more resource intensive and could introduce loss scale issues).

---

> > ### Comment · Reviewer_fwbc · 2024-08-14
> > **Response from Reviewer**
> >
> > I thank the authors for responding to some of the questions I had. Based on their responses regarding clarity and other existing queries I had, I am happy to increase my rating

---

### Official Review · Reviewer_6ptT · 2024-07-12

**Soundness:** 2
**Presentation:** 2
**Contribution:** 2
**Rating:** 5
**Confidence:** 2

**Summary:**

This paper proposes a more efficient method for computing per-example gradient norms without significant computational overhead in terms of FLOPs and I/O cost. This method accurately estimates the gradient noise scale (GNS), useful for neural network batch size selection and scheduling. Additionally, it observes that the normalization layer effectively predicts GNS in transformer models. These findings, along with the proposed algorithm, are applied in a case study on a large language model, where effective batch size scheduling reduces training time.

**Strengths:**

- The proposed method efficiently computes per-example gradient norms and rapidly estimates the transformer gradient noise scale.
- This work also investigates a downstream application of batch size scheduling and demonstrates its time-saving benefits during training.

**Weaknesses:**

- If the proposed method is only supported by heuristic and empirical observations, and if the primary practical use of GNS is to estimate the critical batch size, then the experiment should be much more comprehensive. Only one set of experiments was conducted on a fixed model/dataset, making it unclear if the results are robust without ablation studies. Additionally, the results were not compared to other batch-size scheduling methods.
- The finding that the normalization layer effectively predicts total GNS behaviour in transformers has not been thoroughly tested on other transformers, and it's unclear if this generalizes beyond transformers. The authors also do not provide any intuition as to why this might be the case.
- The paper needs to be more clearly written and easier to read. Some concepts could be better explained or motivated. For example, it's not clear how per-example gradient norms help estimate GNS, what other ingredients are needed, and what other sources of noise there are that should be taken into consideration. It's also challenging to extract key information from the figure captions.

**Questions:**

- Could the author add more experiments on different models and datasets to validate the generalizability of the method? To ensure robustness, it is also important to include ablation studies. Additionally, the results should be compared with other established batch-size scheduling methods to highlight the advantages or disadvantages of the proposed approach.
- In section 2.3, it is claimed that the variance of gradient norms determines whether SGD outperforms adaptive optimizers. However, subsequent studies [Kunstner et al, Noise is not the main factor behind the gap between sgd and adam on transformers, but sign descent might be, ICLR 2023] have shown that the performance gap between Adam and SGD persists even in full-batch scenarios without stochasticity. This suggests that there might not be applications of GNS in this context. It would be beneficial for the author to address this discrepancy.

**Limitations:**

Limitations are well-addressed

---

> ### Author Rebuttal · Authors · 2024-08-07
>
> Thank you for your thoughtful comments and suggestions, and for positively
> noting the efficiency of our GNS estimator.
>
> ### Regarding the scope of the evaluations
>
> > the experiment should be much more comprehensive … only one set of experiments was conducted on a fixed model/dataset, making it unclear if the results are robust without ablation studies.
>
> We agree that sufficient experimental evidence is an important consideration. Of course, in this paper we do not establish the relationship between GNS and critical batch size, as McCandlish et al [32] provided exhaustive experiments on this topic on
> many data and model types.  The focus of our work is validating our novel unbiased estimator of the GNS statistics. To this end, following your suggestion, we will include additional studies for other model types, sizes, and datasets in the revised paper. We propose to expand on Figures 4-6 as follows:
>
> - Figure 4 illustrates the separation of GNS measurements by layer type and Index; this demonstrates the granularity of the estimators in practice.  We have performed the same experiment on many different model types and we will include those results in the camera version of the paper.  We will also add additional plots for other model sizes to the Appendices.
> - Figure 5 is a replication of prior work that opens questions about an experiment from McCandlish's Appendices. We will perform a replication of the original experiment on SVHN in order to check for methodological differences.
> - Figure 6 relates the measurements of GNS across layer types. The results are presented for one model size (111M parameters). We observe this result for language models specifically because that's one large-scale setting where it may be useful. Additional studies for other model types would be valuable here and we will include other model types and datasets in the revised paper.
>
> >  Additionally, the results were not compared to other batch-size scheduling methods.
>
> Note that Figure 8 is presented purely as an example use case for GNS in large-scale language model training. The actual batch size schedule is not novel, and we do not claim to establish a SOTA batch size scheduler.
>
> ### Regarding the generalization of the findings
>
> > The finding that the normalization layer effectively predicts total GNS behaviour in transformers has not been thoroughly tested on other transformers … the authors also do not provide any intuition as to why [the predictability] might be the case … could the author add more experiments on different models and datasets to validate the generalizability?
>
> We agree that it would be interesting to know if this result generalizes to other model types, such as the original post-LayerNorm Transformer, or even to image models, and we will include such studies in the final paper.
>
> Regarding why LayerNorm gradients predict total GNS behaviour, we are currently
> exploring whether the per-example layerwise gradients are correlated for the
> reason that they ultimately reflect variation in the per-example loss. While a full
> theoretical understanding of this phenomenon may prove elusive, we agree that
> some additional experiments to gather greater intuition would be valuable. Thank
> you for this suggestion.
>
> ### Regarding the paper clarity
>
> > Some concepts could be better explained or motivated. For example, it's not clear how per-example gradient norms help estimate GNS... It's also challenging to extract key information from the figure captions.
>
> We regret that Section 2 failed to explain the link between per-example gradient norms and GNS measurements. We suspect the issue may be line 80, where we need to clarify that $B_{big}$ is the full batch and $B_{small}$ are fractions of that same batch.  We will revise section 2 to note that gradient norms are measured for both batch sizes, but typically this can be done cheaply when training DDP because the small batches gradients are on each node and the large batch gradient can be obtained after synchronisation before the update. Per-example gradient norms are intended to obtain a gradient norm for $B_{small}=1$ without requiring any specific training setup (ie DDP). Figure 1 shows that this would give the most accurate measurement of GNS.
>
> We also agree that key information is difficult to extract from Figures. As described in our rebuttal to Reviewer 32m2 we will revise the Figure captions to include the important message of each Figure.
>
> ### Regarding the role of gradient noise in optimizer performance
>
> > In section 2.3, it is claimed that the variance of gradient norms determines whether SGD outperforms adaptive optimizers. However, subsequent studies … have shown that the performance gap persists even in full-batch scenarios without stochasticity
>
> Thank you for referring us to this paper!  Indeed, Kunstner et al. (2023) does strongly suggest that gradient variance is likely not the determining factor for why Adam prevails over SGD in Transformer models.  Our overall point was that our work can support such studies by providing tools to efficiently collect gradient statistics.  However, as you mention, this particular area is perhaps now less well-motivated, and we will revise this part of the related work accordingly.  Thanks again!

---

> ### Comment · Reviewer_6ptT · 2024-08-14
> **Response to authors**
>
> Thank you for addressing my concerns and conducting additional experiments; I will raise my score.

---

### Official Review · Reviewer_5nZ5 · 2024-07-13

**Soundness:** 3
**Presentation:** 3
**Contribution:** 3
**Rating:** 7
**Confidence:** 3

**Summary:**

This work proposes a method to compute per example gradient norms as a means to compute GNS. It shows that not all layers are necessary to estimate the GNS and that the per-example gradient norms can be computed for normalized layers without any overhead.

**Strengths:**

- This work provides an efficient technique for computing the GNS. The technique is supported by various experiments, and elucidates that the GNS of the model is highly correlated between layer types. Furthermore, for LayerNorm layers, this paper develops a custom kernel to compute the backward pass and the per-example gradient, and find that the the throughput overhead of gathering the per-example gradient is 0 (which outperforms PyTorch’s Layernorm).
- The paper also replicates prior GNS observations, which helps support the method.
- I found the experimental results interesting, especially the case study with dynamic batch sizes.

**Weaknesses:**

- As the authors mention, estimating gradient noise scale is useful in training large scale models. The largest model included in the case study only has 111M parameters. I think that this work could benefit from experiments on larger architectures, given the size of language models used in practice today (e.g., at least 7B).
- This study is limited to transformers. However, many other architectures do not use normalization sub-layers. This point is also acknowledged by the authors.

**Questions:**

Perhaps the work can benefit from experiments with larger model sizes, given that even small open source models have 7B parameters.

**Limitations:**

As acknowledged by the authors, this paper is limited to transformers, which inherently use normalization sublayer (other architectures do not conventionally use such layers).

---

> ### Author Rebuttal · Authors · 2024-08-07
>
> Thank you very much for your thoughtful feedback, and for your support of the paper's core idea and experimental approach.
>
> ### Regarding larger-scale models
>
> > The largest model included in the case study only has 111M parameters... this work could benefit from experiments on larger architectures
>
> Yes, we agree that demonstrating our findings on larger models would be
> beneficial.  Following your suggestion, we have conducted experiments on a 1.3B
> parameter model. The results are in our attached pdf document in the response
> to all reviewers.
>
> ### Regarding architectures beyond transformers
>
> > This study is limited to transformers. However, many other architectures do not use normalization sub-layers. This point is also acknowledged by the authors.
>
> We agree that this is a limitation.  However, we should have also mentioned in
> the paper that our per-example gradient estimation methods for other layers
> still apply (Algorithms 1 and 3). Although perhaps not as performant as
> gathering statistics from the LayerNorms -- depending on the kernel used -- the
> increase in runtime may be acceptable. In experiments we have run it was at
> worst 30% slower to gather per-example gradient norms for all layers. Gathering
> only LayerNorm per-example gradients did not slow down training at all. We will
> revise the paper to make this observation. Thanks!

---

> > ### Comment · Reviewer_5nZ5 · 2024-08-12
> >
> > I thank the authors for their response and maintain my score.

---

### Author Rebuttal · Authors · 2024-08-07

Thank you to all the reviewers for their thoughtful feedback. After reading the
reviews we noted the following points we could address with additional figures:

1. Reviewers 6ptT, fwbc and 32m2 brought attention to the clarity of the work.
2. Reviewers 5nZ5 and 6ptT asked for additional experiments on other models or
   datasets.
3. Reviewer 32m2 asked about the numerical stability, restrictions to DDP and
   requirements for gradient accumulation.

To address the first point we provide Figure 1 in our attached document. This
figure presents a diagram to explain some of the key concepts, ie that we are
dealing with gradient norms that vary across minibatches and between layers.
Additionally, it presents an intuition for why the gradient norms between layer
types may be similar.

To address the second point we provide Figures 2a and 2b in our attached document.
This figure illustrates the results of a 1.3B GPT model trained on
OpenWebText twice from scratch on 8 H100 GPUs. Both runs were configured to
match the 1.3B Chinchilla optimal Cerebras-GPT training run. The GNS was also
gathered at the same time in both runs using the traditional DDP method (small batch gradient
norms gathered on each node). The runs were numerically stable after addressing
issues independent to our method (details in response to Reviewer 32m2).

The first run gathered per-example gradient norms for all layers and ran at 40%
[MFU][] (p.9). Figure 2a repeats the analysis of Figure 6 in the paper, finding
again that the total GNS is well predicted by only the LayerNorm per-example
gradient norms. However, it was found that the slope of the regression is higher
at approximately 2.2. This indicates that it may be necessary to calibrate the
estimate of the GNS intermittently by enabling all per-example gradient norms
for a short time during training, or matching it to the GNS gathered by DDP, if
available.

The second run enabled only the LayerNorm per-example gradient norms and ran at
57% [MFU][]. Using the 2.2x calibration factor the GNS is plotted against the GNS
gathered at the same time by DDP in Figure 2b. The GNS estimates are very close
throughout training. This would allow, for example, continued tracking of the
GNS if we were to move the run to one GPU during training, at which point
tracking the GNS via DDP would not be possible.

[mfu]: https://arxiv.org/abs/2204.02311

### On numerical stability

Thanks to reviewer 32m2 for raising an important point about the numerical
stability of the work that we would like to address to all reviewers to make
sure it is clear that our method does not affect the numerical stability of
training.

> Some more work is required to make the proposed method practical due to numerical issues. Although the idea is interesting it would be much more impactful if the kernel worked as a drop-in replacement.

The numerical issues mentioned in the paper are indeed an issue we spent a lot
of time on. However, they are not due to our method; early in testing we
disabled our code and found the same issues on the main branch of nanoGPT.

Specifically, the following config is sufficient to reproduce the issue:
```
batch_size=8
gradient_accumulation_steps=2
block_size=2048
max_iters=72_000
lr_decay_iters=72_000
warmup_iters=1000
```
It is caused by an interaction with flash attention and AMP in PyTorch.
Disabling flash attention or training float32 resolves it. Some other users have
made issues documenting similar behaviour in the nanoGPT repository, such as
[here](https://github.com/karpathy/nanoGPT/issues/137). We have since resolved
these issues with small architectural changes.

The kernel does work as a drop in replacement and we agree that this is vital.
We hope additional experiments shared in our attached pdf at 1.3B scale on
a distributed setup should make this clear. Complete code will be released with
the camera-ready version of the paper.

> The 18% speed improvement claimed in the abstract may be an overclaim due to the numerical issues and presumably only being applicable to certain types of training setups (maybe one GPU doing gradient accumulation rather than distributed setups).

This is an important concern. As shown in our attached rebuttal pdf, we are able
to train DDP at large scale with this method. Additionally, it does not depend
on gradient accumulation, although that is another way to gather small batch
gradient norms.

---

### Decision · Program_Chairs · 2024-09-25

**Decision:**

Accept (poster)

**Comment:**

This paper provides observations on gradient norms, which lead to removal of some nearly-redundant calculations across layers and thus speedups.  Reviewers had a variety of concerns regarding the rigor and presentation quality, but overall the paper is above bar and I am happy to recommend acceptance.